# Flexible Sensing Enabled Nondestructive Detection on Viability/Quality of Live Edible Oyster

**DOI:** 10.3390/foods13010167

**Published:** 2024-01-03

**Authors:** Pengfei Liu, Xiaotian Qu, Xiaoshuan Zhang, Ruiqin Ma

**Affiliations:** College of Engineering, China Agricultural University, Beijing 100083, China; liu_pengfei@cau.edu.cn (P.L.); quxiaotian@cau.edu.cn (X.Q.); zhxshuan@cau.edu.cn (X.Z.)

**Keywords:** live oysters, HACCP system, flexible wireless sensors networks, quality control

## Abstract

Environmental and physiological fluctuations in the live oyster cold chain can result in reduced survival and quality. In this study, a flexible wireless sensor network (F-WSN) monitoring system combined with knowledge engineering was designed and developed to monitor environmental information and physiological fluctuations in the live oyster cold chain. Based on the Hazard Analysis and Critical Control Point (HACCP) plan to identify the critical control points (CCPs) in the live oyster cold chain, the F-WSN was utilized to conduct tracking and collection experiments in real scenarios from Yantai, Shandong Province, to Beijing. The knowledge model for shelf-life and quality prediction based on environmental information and physiological fluctuations was established, and the prediction accuracies of TVB-N, TVC, and pH were 96%, 85%, and 97%, respectively, and the prediction accuracy of viability was 96%. Relevant managers, workers, and experts were invited to participate in the efficiency and applicability assessment of the established system. The results indicated that combining F-WSN monitoring with knowledge-based HACCP modeling is an effective approach to improving the transparency of cold chain management, reducing quality and safety risks in the oyster industry, and promoting the sharing and reuse of HACCP knowledge in the oyster cold chain.

## 1. Introduction

An oyster is a kind of nutrient-rich bivalve mollusk; it is, rich in liver glycogen, protein, vitamins, fatty acids, and other nutrients [1,2,3,4]. At the same time, live oysters are more popular with consumers in China as they maintain optimal flavor and conform to dietary habits [5,6,7]. Therefore, live oyster marketing is regarded as a value-adding process because live oysters obtain substantially higher prices with lower processing costs. The cold chain of live oysters effectively preserves their nutritional value and improves economic benefits, providing a green alternative. Nevertheless, live oysters are a fragile and highly perishable product that is easily affected by acute stress during the cold chain. Meanwhile, transporting large quantities of live oysters is a relatively complex process, which requires better control of potential transport stress in order to prevent physiological stress responses to oysters [8]. Additionally, prolonged exposure to unsuitable environments can cause damage to the oyster’s tissues and reduce their survival ability [9]. These factors can work together and lead to continuous physiological stress responses and even result in the death of the oysters [10,11,12]. Therefore, it is urgent to establish a comprehensive traceability system to improve the quality control and transportation management of live oysters, thus ensuring that higher quality oysters are guaranteed in the market [13].

Hazard Analysis and Critical Control Points (HACCPs) is an effective quality control system widely used in the food industry, especially for meat, fish, and shellfish, aimed at reducing product safety risks [14,15]. However, the effective implementation of HACCP relies not only on practical experience and knowledge but also on continuous monitoring measures and instant feedback during the actual cold chain process. At present, the general HACCP system is based on historical static knowledge and lacks process control and dynamic adjustment of the HACCP plan, which leads to the following deficiencies in the actual transportation and application of raw oysters:The HACCP system does not form a unified, reusable business of knowledge in the live oyster cold chain.The current HACCP system focuses on risk prevention, and the risks in the process cannot be identified and resolved in time. However, dynamic changes such as temperature, humidity, and physiology require continuous monitoring and dynamic adjustment [16].The implementation of HACCP requires specialized knowledge and skills. However, the traditional HACCP system is manually operated, lacking complete records and a decision support system (DSS), which is not only time-consuming but also inefficient.

With the rise of wireless transmission technologies such as Zig Bee, 6LoWPAN, and IEEE 802.15.4 [17,18,19,20], wireless sensor networks (WSNs) are commonly used for the real-time collection of environmental monitoring indicators. In our previous study, WSNs were effectively used for monitoring physiological stress signals during the waterless transport of live fish [21]. Obviously, the monitoring and control of critical indicators is necessary for the early warning of the HACCP system in the cold chain, which can improve the terminal quality of products. However, the traditional wireless sensor network has the disadvantages of rigidity, large volume, and high power consumption, and there is no research on tracking the live oyster cold chain. Developing low-cost, flexible, low-power, and advanced network capabilities and establishing uniform and effective knowledge rules by selecting critical control points (CCPs) is an important challenge. Therefore, it is necessary to rebuild the HACCP quality control system to realize dynamic adjustment and process control to improve the quality and safety of oysters [22].

Knowledge modeling is an important research topic in the field of the Internet of Things (IoT), which focuses on how to effectively represent and reason about knowledge to promote knowledge sharing and reuse. In the live oyster cold chain, there are many challenges in ensuring the integrity and consistency of HACCP knowledge sharing and transfer. Therefore, it is necessary to establish a systematic HACCP knowledge modeling framework to ensure the consistency, accuracy, and sharing of knowledge reasoning rules to improve cold chain efficiency in implementation.

The objective of this study was to improve the quality control of live oysters using a flexible wireless sensor network (F-WSN) monitoring system and knowledge-based HACCP modeling. In this paper, the Yantai Rushan oyster was used as the experimental research object, and a wireless sensor acquisition device that is flexible, biocompatible, and suitable for oyster bodies is designed and fabricated. An F-WSN real-time monitoring system was used to obtain microenvironment and physiological parameters of the live oyster cold chain. Moreover, this research integrated Internet of Things (IoT) and artificial intelligence (AI) models to investigate the interplay between several quality indicators (TVB-N, TVC, and pH) and the coupling relationship with physiological stress and microenvironment. The unified knowledge representation and inference rules of the live oyster HACCP system were established. This study contributed to live oyster HACCP knowledge sharing and transfer and improved cold chain management efficiency.

## 2. Materials and Methods

### 2.1. Conceptual Framework for Quality Control of Live Oysters

Improper handling in the cold chain exposes live oysters to chronic stress that affects the quality of the product at the cold chain terminal [23]. It is necessary to strengthen effective quality control measures to ensure product quality in this scenario. Combining flexible wireless sensor networks (F-WSNs) with decision-support Hazard Analysis and Critical Control Point (HACCP) systems to monitor the entire production process of live oysters in real-time can improve quality control and decision-making capabilities, thereby reducing major risks and quality issues during the production process [24,25].

In conclusion, the quality characteristics of the final product are determined by considering various influencing factors. The oyster cold chain critical control points (CCPs) include physiological fluctuations and microenvironment indicators. Therefore, real-time monitoring, data acquisition, and transmission of physiological fluctuations and microenvironment indicators through F-WSN have become important requirements for obtaining dynamic quality information. We established corresponding critical limit values for each CCP that had been determined and used F-WSN to identify CCPs and assess potential hazards for the HACCP plan. When the CCP data are found to deviate from the preset limit value in WSN collection and transmission of dynamic data, the system gives a timely alarm and takes control measures to prevent unqualified products from reaching consumers. Meanwhile, the traceability system is combined with the knowledge base and Internet of Things (IoT) technology to assist users in effectively managing and solving live oyster cold chain quality problems. Figure 1 demonstrates the conceptual framework for live oyster cold chain quality control. As can be seen from Figure 1, the live oyster quality control supply chain includes three parts: microenvironment parameters, potential danger signals, and the HACCP system. The physical, biological, and chemical signals of potential hazards will be directly transmitted to the knowledge database, which will affect the prediction of survival, shelf-life, and quality. The fluctuation of glycogen, lactic acid, nutrition, protein, and enzyme activity in the HACCP system will directly affect the quality, health status, and survival rate of oysters, and also affect their physiological characteristics. Similarly, microenvironment parameters such as temperature, humidity, and ammonia also directly affect the quality, health status, and survival rate of oysters. Microenvironment parameters and potential hazards are transmitted to monitoring procedures and monitoring systems to influence decision support systems. Survival prediction, shelf-life prediction, quality prediction, and decision support work together to improve the quality management of live oysters.

### 2.2. Architectural Design of a Flexible Wireless Sensor Network (F-WSN) Monitoring System

The live oyster cold chain F-WSN monitoring system includes 5 layers: the F-WSN data layer, network layer, database layer, application layer, and visual display layer. Figure 2 shows the flexible wireless sensor network (F-WSN) system architecture.
The data layer consists of wireless sensor nodes distributed in the monitoring area to cooperatively form the target perception field. It is responsible for collecting microenvironmental information (temperature, humidity) and physiological information (shell-closing strength, heart rate) in the live oyster cold chain.The network layer is the link between the receiving and sending convergence point and the user, ensuring that the target signal of the sensor node is transmitted through the Internet and satellite communication after simple processing, collection, and aggregation.The database layer mainly includes a database, data warehouse, algorithm library, and expert knowledge base. The database is responsible for collecting microenvironmental and physiological signals monitored by F-WSN transmitted in real-time. The data warehouse organizes the data collected by the F-WSN to provide useful guidance for decision-making. The algorithm library is responsible for storing kinetic model equations, including live oyster survival rate prediction models and quality prediction models. The expert knowledge base is mainly responsible for the storage and algorithm library, verifying the reasoning rules of the HACCP system, statistically analyzing the monitoring content of the CCPs, and adopting corresponding corrective measures.The application layer uses the Internet of Things (IoT) protocol to communicate with the F-WSN device and standardizes data transmission through the protocol format. Users interact with the F-WSN through data transmission and analysis of various smart devices.The presentation layer provides users with a visual environment and a graphical user interface (GUI), providing real-time data to headquarters, managers, field workers, and experts. The manager interface is used to view the dynamic change graph of real-time data and the processing results of the algorithm library.

### 2.3. Knowledge Structure Analysis and Construction of Live Oyster HACCP System

The HACCP system as a scientific method to assess the hazards associated with food production could effectively control food safety. Therefore, this study integrates expert opinion and knowledge base to establish a HACCP plan for better management of live oyster cold chains. As shown in Figure 3, the HACCP plan for the live oyster cold chain consists of the following steps:(1)Identify and assess potential hazards at each stage of the live oyster cold chain and conduct analysis, estimate the probability of potential hazards occurring, and determine preventive measures.(2)Identify and control critical control points (CCPs) in the live oyster cold chain.(3)Confirm critical limits (CLs) for live oyster quality and ensure that critical control points are under control.(4)Establish a wireless sensor network and evaluation system based on the HACCP system to monitor the microenvironment and physiological signals of oysters. By comparing the results of the determined critical control points (CCPs) with the critical limit (CL), whether the critical point is effectively controlled is determined.(5)When the monitoring system finds that a critical control point (CCP) is out of control, corrective measures are taken in time.(6)Establish effective record storage procedures to ensure the effective operation of the HACCP system.(7)Establish a live oyster cold chain process documentation system in compliance with HACCP principles, and verify the normal operation and revision records of the HACCP system.

### 2.4. HACCP System Knowledge Modeling

#### 2.4.1. Vitality Evaluation Knowledge Model

We applied the HACCP system through effective methods to obtain the optimal transport temperature to ensure the survival rate of oysters. The critical factors affecting oyster survival include environmental indicators, stress levels, and metabolic consumption [26]. When given the transport time or distance, mathematical models could be constructed to dynamically predict the survival rate of the terminal. The error values are continuously optimized for the collected samples using Equation (1) to obtain the optimal temperature for maintaining normal metabolism.
(1)$$Temp_{loss}=\overset{n}{\underset{i=1}{\sum }}\frac{{\left(Temp_{prei}-Temp_{acti}\right)}^{2}}{2}$$
where $Temp_{prei}$ is the predicted optimal temperature, $Temp_{acti}$ is the actual temperature at which the samples were transported, and $Temp_{loss}$ is the temperature error value.

The F-WSN can efficiently statistically calculate each influence factor and determine the optimal parameters for each factor using the method described above. Transport times are arranged according to a determined time interval $\left(h\right)$, and survival rates are predicted using a time series model with one exponential smoothing method. The smooth recursive relationship of the survival rate is as follows from Formula (2):(2)$$Survival_{pre|i}=\delta Survival_{act|i}+\left(1-\delta \right)Survival_{rate|i-1}$$

$Survival_{rate|i}$ is the smoothed value at time node i, and $Survival_{act|i}$ is the actual survival rate currently node. δ can be any value between 0 and 1, and it controls the balance between the survival rate at the previous time node and the next node: when δ is close to 1, only the current survival rate data point is retained; when δ is close to 0, only the previous smoothed value is retained. The detailed recursive relational equation is as follows:(3)$$Survival_{ratei}=\delta \overset{i}{\underset{j=0}{\sum }}\left(1-\delta \right)Survival_{act|i-j}$$

#### 2.4.2. Quality Evaluation Knowledge Model

As the quality of live oysters decreases, their physiological signals gradually undergo a series of fluctuations. The data obtained from the wireless sensor network can be collected and fused into a back propagation artificial neural network (BP-ANN) for quality assessment. The BP-ANN is a multilayer feedforward network with back propagation through error. It is a gradient descent method to minimize the mean square error between the actual output value of the network and the desired output value [27]. A BP-ANN usually consists of three layers: the input layer, the implicit layer, and the output layer. When the wireless sensor network acquires environmental and physiological parameters, the threshold and weights are adjusted layer by layer by a gradient descent strategy to obtain a neural network model that can effectively evaluate the quality. The specific steps of the quality assessment model are as follows:

Step 1: Calculate the output of each node sequentially according to the oyster sample parameters $\left\{x_{1},x_{2},x_{3}\ldots x_{n}\right\}$:(4)$$h_{j}^{in}=\overset{n}{\underset{i=1}{\sum }}W_{ij}^{\left(1\right)}x_{i},\;h_{j}^{out}=f\left(h_{j}^{in}\right),j=1,2,\ldots ,p$$
(5)$$h_{k}^{in}=\overset{p}{\underset{j=1}{\sum }}W_{jk}^{\left(2\right)}h_{j}^{out},\;y_{k}^{out}=f\left(y_{k}^{in}\right),k=1,2,\ldots ,m$$
(6)$$L=\frac{1}{2}{\left\|y^{out}-y^{l}\right\|}^{2}$$
where (3) is the input and output of the hidden layer, (4) is the input and output of the output layer, and (5) is the calculated loss function.

Step 2: Calculate the gradient of each node according to the loss function:(7)$$\frac{\partial L}{\partial y_{k}^{in}}=\left(y_{k}^{out}-y_{k}^{l}\right)\cdot y_{k}^{out}\cdot \left(1-y_{k}^{out}\right)$$
(8)$$\frac{\partial L}{\partial W_{jk}^{\left(2\right)}}=\left(y_{k}^{out}-y_{k}^{l}\right)\cdot y_{k}^{out}\cdot \left(1-y_{k}^{out}\right)\cdot y_{j}^{out}$$
(9)$$\frac{\partial L}{\partial W_{ij}^{\left(1\right)}}=\frac{\partial L}{\partial h_{j}^{out}}\cdot \frac{\partial h_{j}^{out}}{\partial h_{j}^{in}}\cdot \frac{\partial \partial h_{j}^{in}}{\partial W_{ij}^{\left(1\right)}}$$

Step 3: Update the parameters using gradient descent algorithm:(10)$$W_{ij}^{\left(1\right)}=W_{ij}^{\left(1\right)}-\frac{\partial L}{\partial W_{ij}^{\left(1\right)}}$$
(11)$$W_{jk}^{\left(2\right)}=W_{jk}^{\left(2\right)}-\frac{\partial L}{\partial W_{jk}^{\left(2\right)}}$$

The output and input functions of neurons in each implicit layer and output layer are related as Equation (12):(12)$$I_{j}=\sum_{i}W_{ij}O_{i}$$
(13)$$O_{i}=sigmoid\left(I_{l}\right)=\frac{1}{1+e^{-I_{l}}}$$

### 2.5. HACCP Quality Control System Evaluation

The purpose of the system evaluation is to verify the effectiveness, efficiency, and reliability of the live oyster HACCP system based on F-WSN, as well as the advantages of oyster survival and quality. The results of the evaluation are used to improve the efficiency of making decisions for improvement and optimization to meet the needs and objectives of enterprises for live oyster transportation. In this study, the HACCP system evaluation passed three aspects to analyze three aspects: (1) analysis of system design, operation, and user needs; (2) testing and analysis of the safety, reliability, performance, and scalability of the system; and (3) related manager, employees, and experts are invited to participate in system efficiency analysis and system quality assessment. System efficiency analysis and quality assessment are based on the questionnaire survey scores. The subjective evaluation of these invited reviewers and comparison with previous research results. It should be noted that the questionnaire survey uses a 5-point standard: 1 point represents the worst performance and 5 points represent the best performance. The final evaluation analysis results will be displayed in the evaluation table. Through systematic evaluation, it can effectively help enterprise managers understand the current situation of its system and provide opinions and suggestions on the improvement plan of the live oyster HACCP system.

### 2.6. Experimental Scenario Analysis and Implementation

The experimental material, Pacific oysters, had a mean weight (±SD) of 300 ± 10 g with a clean appearance and no damage. After harvesting, the oysters were temporarily reared at 4 °C for 48 h in a temporary pond. These live oysters were packed in special transport boxes equipped with intelligent sensor systems. In this study, business process modeling technology was used to track the transportation process of the entire live oyster cold chain, and harmful analysis was carried out at each transportation stage to effectively obtain critical control points (CCPs). The process of determining critical control points (CCPs) of oyster cold chain transportation can be divided into the following steps:

Step 1: Live oyster farming and harvesting. Oysters were raised by farmers or workers on farms. The breeding environment was required to be safe and free from metal pollution and water pollution.

Step 2: Grading and temporary holding. The caught live oysters were graded according to weight and kept in temporary rearing tanks for 1–2 days to adapt to the new environment. During this period, the water temperature was kept at 2–4 °C.

Step 3: Clean surface. Special cleaning machines were used to remove dirt from the surface of the oysters to ensure that they were clean during transport.

Step 4: Packaging. Oysters were transported in a specially designed transport box that automatically controlled and adjusted environmental parameters. At the same time, a sensor monitoring device was installed on individual oysters to ensure that physiological signals of oysters were obtained throughout the process.

Step 5: Transportation. The transport of live oysters was a crucial part of the process. In this process, live oysters were placed in transport boxes and then loaded onto transport trucks for transportation. The temperature of the truck was kept at 4 °C to ensure optimum temperature for the oysters. In addition, the F-WSN intelligent system monitored environmental parameters such as temperature and humidity as well as physiological parameters in real-time and enabled automatic regulation and control.

Step 6: Storage. Oysters were transported to a warehouse for storage at 4 °C.

Step 7: Market sales. In this step, wholesalers, retailers, and customers were jointly responsible. Once the oysters were evaluated, they were delivered to the next distributor. The oysters were classified and sold based on their individual survival and quality.

The oysters were transported from Yantai City, Shandong Province, China to Haidian District, Beijing, China, and stored after transport to the destination. At the same time, simulated transportation was carried out on the cold chain transportation test bench as the control group. The whole experimental scenario mainly includes two aspects: a live oyster transportation test and a biochemical test. In the biochemical test, the colony count was measured by a plate coating method; volatile salt-based nitrogen was measured by the Kjeldahl method; and the pH was measured by a special pH meter. The business process for the live oyster cold chain is shown in Figure 4. The general process of the oyster cold chain business is as follows: after harvesting, grading, and temporary feeding follows cleaning the surface, installing environmental sensors, packaging and transportation, using sensors to monitor their physiology during transportation, and, finally, storing and marketing.

## 3. Results and Discussion

### 3.1. Planning of Quality Management of Live Oyster Cold Chain

The establishment of the HACCP system in the live oyster cold chain can prevent, eliminate, or reduce quality safety issues to an acceptable level. In this study, the live oyster cold chain was followed throughout the cold chain and risk analyses of the HACCP system were conducted. On this basis, this study proposes a knowledge-based cold chain management scheme for live oysters aimed at achieving the goal of dynamic quality adjustment. The use of knowledge-based management schemes could provide targeted solutions based on previous research. The oyster cold chain quality management planning and implementation scheme is shown in Table 1.

HACCP domain knowledge can be abstracted into a knowledge representation model [28]. The oyster cold chain HACCP knowledge domain consists of a set of abstract concepts $\left(C\right)$ and a set of instances $\left(I\right)$ associated with those concepts [29]. The concept set is a collection of all the concepts included in the HACCP plan, $C_{n}=\left\{C_{1},C_{2},©,C_{i}\right\}$; each concept contains multiple instance sets, $I_{n}=\left\{I_{1},I_{2},©,I_{i}\right\}$, where $n=\left\{1,2,©,i\right\}$. The HACCP unified expression model based on knowledge is defined as follows:$$\mathrm{HACCP}\;=\left\{\mathrm{CCP},\;\mathrm{HA},\;\mathrm{CM},\;\mathrm{CL},\;\mathrm{CA},\;\mathrm{VE},\;\mathrm{RE}\right\}$$
where $Si=\left[S1,\;S2,\;©Sn\right]$ represents all steps in the live oyster cold chain; CCPs means critical control points; HA means hazard analysis; CL means critical limit; CM means critical monitoring content; CA means corrective action; VE means verification; RE means record.

$\mathrm{CCP}=\left[\mathrm{CCP}1,\;\mathrm{CCP}2,\;©\mathrm{CCPi}\right]$ indicates the set of critical control points in the live oyster cold chain;

$\mathrm{HA}=\left[\mathrm{HA}1,\mathrm{HA}2,\;©\mathrm{HAi}\right]$ indicates the set of potential hazards in the live oyster cold chain;

$\mathrm{CM}=\left[\mathrm{CM}1,\mathrm{CM}2,\;©\mathrm{CMi}\right]$ indicates the set of critical monitoring content for potential hazards;

$\mathrm{CL}=\left[\mathrm{CL}1,\;\mathrm{CL}2,\;©\mathrm{CLi}\right]$ indicates the set of critical limits for potential hazards;

$\mathrm{CA}=\left[\mathrm{CA}1,\;\mathrm{CA}2,\;©\mathrm{CAi}\right]$ indicates the set of corrective action; 

$\mathrm{VE}=\left[\mathrm{VE}1,\;\mathrm{VE}2,\;©\mathrm{VEi}\right]$ indicates the set of verification results for CCPs; 

$\mathrm{RE}=\left[\mathrm{KP}1,\;\mathrm{KP}2,\;©\mathrm{KPi}\right]$ indicates the set of records for HACCP system.

### 3.2. Physiological and Environmental Signal Acquisition at Critical Control Points

Information about the environment and physiology captured by the flexible wireless sensor network (F-WSN) provides valid data for the HACCP system. The knowledge about each critical control point (CCP), critical limit (CL), corrective action, and validation in the HACCP plan is obtained using the data acquired by the F-WSN. The data captured by the F-WSN is shown in Figure 5. Figure 5 mainly shows the changes in temperature, humidity, shell strength, and heart rate of oysters during the whole stage from harvest to market sale.

#### 3.2.1. Temperature Variation in Cold Chain

In this study, the dynamic temperature of the cold chain was analyzed to evaluate the influence of the environment on the quality of live oysters at each stage. The F-WSN collected the temperature and humidity of 14 days in the actual scene and analyzed the change trend by using the stage analysis method. Temperature changes in oyster cold chain are shown in Figure 5a. In Figure 5a, the X-axis represents time, the Y-axis represents the five stages of fishing, temporary rearing, transportation, storage, and marketing, and the Z-axis represents the temperature fluctuation in different stages. At the same time, this study further analyzed the temperature and duration of each stage in the entire scenario, as shown in Table 2. Firstly, the oysters were loaded into the truck after harvesting at the farm, and the harvest temperature was 17.86 °C for 2 h. The temperature was regulated during the temporary rearing stage, decreasing from the initial 17.5 °C to 4.02 °C. In the clean surface stage, the workshop temperature was 4.08 °C for 3 h. In the next step, the live oysters were loaded into the special transportation box and transported from Yantai to Beijing through low-temperature transportation. The initial temperature in the cold chain truck was 17.6 °C, the temperature dropped to about 4 °C after 48 min, and the whole transportation time was 13 h. In the storage stage, oysters were kept in cold storage at an average temperature of 4.01 °C for 7 days. In the final marketing stage, the live oysters were placed in a supermarket fresh-keeping cabinet at 7.04 °C for 1 d. The statistical results demonstrated that temperature fluctuations at different stages of the cold chain are influenced by multidimensional factors [29]. Therefore, the adjustment of temperature in the cold chain is crucial to ensure the quality of live oysters [30].

#### 3.2.2. Humidity Variation in Cold Chain

Humidity monitoring is critical in the cold chain to ensure the quality of oysters, and the fluctuations in humidity throughout the process are shown in Figure 5b. In Figure 5b, the X-axis represents time, the Y-axis represents the five stages of fishing, temporary rearing, transportation, storage, and marketing, and the Z-axis represents the humidity fluctuation in different stages. In the harvesting stage, the variation ranged from 58.1% to 67.7% due to the difference in humidity between day and night. In the temporary rearing stage, the humidity fluctuated from 67.9% to 92.2%. In the transport stage, the humidity fluctuated less from the initial 64% and at the end reduced to 61%. During the storage phase, the humidity was relatively stable and remained around 60%. During the marketing phase, the oysters were placed on the shelves and the humidity ranged from 87.2% to 89.6%. The experimental results indicated that the humidity fluctuated widely in the cold chain and could not meet the requirement of maintaining the oysters in optimal condition. Therefore, the effective regulation of humidity, which is the most important factor affecting the survival and quality of oysters, has become an urgent issue [31].

#### 3.2.3. Shell-Closing Strength Variation of Live Oysters

Oysters were subjected to factors such as external stress (vibration, temperature, and light) after harvesting, and the shells were kept tightly closed to protect against external risks. The shell-closing strength (SCS) of oysters decreased gradually over time, and this process was divided into three periods. The trend of live oyster SCS over time is shown in Figure 5c and Table 3. In Figure 5c, the X-axis represents the change in temperature, the Y-axis represents the change in humidity, and the Z-axis represents the change of shell-closing strength (SCS) under the change of temperature and humidity. In the first period (0–5 days), the SCS of live oysters decreased slowly, i.e., from the initial 3511 g to 3209 g on the 5th day. This may have been due to the storage of nutrients such as glycogen in the oyster body, which ensured that the shell of the oyster was maintained in a high state of closure after external stress. In the second period (6–12 days), the nutrients such as glycogen stored in oysters were gradually consumed, resulting in a gradual acceleration of the decline rate of SCS from 3029 to 1359 g. This indicated that the vitality of the oysters gradually weakened, causing the SCS to be unable to maintain the previously tight closure state [26]. In the third period (13–14 days), the SCS decreased rapidly from 1359 to 0. This was because of the significant loss of moisture in the oyster’s body, accompanied by the abundant growth of bacteria, which led to the rapid weakening of the oyster’s vitality until eventual death [32]. Additionally, this may have also been related to the gradual deterioration of the physiological state of the oysters after experiencing external stress [33].

#### 3.2.4. Heart Rate Variation of Live Oysters

Heart rate is an important physiological indicator reflecting the organism’s state in the oyster cold chain. The heart rate of oysters under different temperature and humidity conditions is shown in Figure 5d. In Figure 5d, the X-axis represents the change in temperature, the Y-axis represents the change in humidity, and the Z-axis represents the change in heart rate (HR) under different temperature and humidity changes. During the whole cold chain, the heart rate fluctuation of oysters ranged from 17 bpm to 32 bpm and was mainly between 20 bpm and 30 bpm. The projection of space points on the A-side showed that there were significant differences in the heart rate of oysters at different temperatures (*p* < 0.01). This was probably because temperature affected oyster metabolism, which was more vigorous at higher temperatures [2]. The projection of spatial points on the B-side showed no significant difference (*p* < 0.01) in heart rates at different humidity. The heart rate of live oysters in different environments is shown in Figure 5d.

### 3.3. Dynamic Quality Adjustment Scheme of Live Oyster Cold Chain Transportation

According to the quality control plan, it is necessary to conduct real-time monitoring of the critical control points throughout the entire cold chain. This enables the identification of any instances where the critical limits are exceeded and facilitates appropriate adjustments. At the same time, the establishment of quality control real-time monitoring measures to confirm the effectiveness of the HACCP plan. In the cold chain of live oysters, physiological and environmental information collected by the F-WSN was integrated and transmitted to the decision support system. We compared the monitoring results of identified critical control points (CCPs) with the critical limits to determine whether the critical control points were effectively controlled. Following the HACCP control plan, the decision support system determines the current status of the oyster and adjusts the CCPs.

The HACCP dynamic adjustment plan mainly includes two aspects: on the one hand, critical control point identification and analysis, including potential risk factors, preventive measures, critical limits, and harmful consequences; on the other hand, monitoring measures, including monitoring targets, monitoring thresholds, and feedback control [28]. Cold chain critical control point adjustments include the following: CCP1 farming and fishing, CCP2 grading and temporary rearing, CCP3 clean surfaces, CCP4 packaging, CCP5 transportation, and CCP6 market sales. The adjustment plan is based on the physiological signals of the oysters and the results of the quality evaluation. The knowledge-based critical control point adjustment plan is shown in Figure 6.

### 3.4. Knowledge-Based Modeling for Oyster Shelf-Life and Quality Evaluation

Knowledge modeling is a key method to effectively store, utilize, and manage the data collected by wireless sensor network (F-WSN) systems [29]. It uses the data obtained by F-WSN to obtain relevant knowledge for each critical control point (CCP), critical limit (CL), corrective action, and validation in the HACCP plan. The F-WSN monitors microenvironmental information and oyster physiological signals, and the knowledge-based model is used to effectively assess the shelf-life and quality of live oysters. If the monitored value does not meet the requirement of CL, the corrective action of rejection will be taken. At the same time, the evaluation results need to be verified and corrected [34]. Regardless of whether the monitoring results meet the minimum requirements, a record should be made to provide a basis for acceptance or rejection of live oysters.

#### 3.4.1. Vitality Evaluation of Oyster Cold Chain Based on the Knowledge Model

In the comprehensive evaluation of the effectiveness of the cold chain system, ensuring the survival of oysters at the end of the cold chain is an important indicator. In this study, 70% of oysters were selected as the training set, 15% as the test set, and 15% as the validation set. The results of the shelf-life prediction model are shown in Figure 7. In the prediction results, the MSE = 0.9239, RMSE = 0.9612, and coefficient of determination = 0.96, indicating that the prediction model can evaluate the survival rate of live oysters during the whole transportation process in real-time. This study can further explore how to improve the survival rate of oysters by optimizing transport, handling, and storage. In addition, relevant indicators and model optimization, such as biological models, can be considered to assess and predict the survival of terminal oysters [35]. This will provide more in-depth theoretical support and practical guidance for improving the sustainable development of cold chain systems and oyster quality assurance. 

#### 3.4.2. Quality Evaluation of Oyster Cold Chain Based on the Knowledge Model

The system knowledge base effectively stores and organizes oyster information to support decision-making and operations during the implementation and operation of the HACCP system [36]. Unreasonable practices and critical control points in the cold chain could cause physiological stress and irreversible damage to oysters. Adopting reasonable methods to assess oyster quality for effective regulation is the most important issue. Figure 8 mainly shows the changes of the TVB-N, TVC, and pH of oysters during the whole stage from harvesting to marketing. The X-axis represents time, the Y-axis represents temperature change, and the Z-axis represents the change of quality indicators (TVB-N, TVC, and pH) under time and temperature change.

Deterioration in the quality of live oysters over time occurs throughout the cold chain process [37]. TVB-N, TVC, and pH were selected as quality evaluation indicators to explore the quality deterioration process of oysters. The initial content of TVB-N in live oysters was 2.01, and it gradually increased over time, reaching a maximum of 7.83 on the 14th day, as shown in Figure 8a. The lack of effective regulation of CCPs during the harvesting, storage, transport, and marketing of live oysters potentially leads to increased levels of TVC in oysters. The initial TVC content is 2.08, which gradually increases over time and reaches 5.25 on the 14th day, as shown in Figure 8b. Therefore, proper hygienic handling procedures are important measures to control the microbiological population in oysters and to ensure food safety [38]. The initial pH value is 7, but it gradually decreases over time, reaching 6.07 on the 14th day, as shown in Figure 8c. This is due to the dissolution of carbon dioxide produced by the metabolism of live oysters in the tissues leading to a decrease in pH. In addition, decay and bacterial decomposition are also important causes of pH reduction [39].

In this study, live oyster quality was assessed in real-time using a machine-learning model. At the same time, the performance of the model is evaluated by comparing the simulation results with the actual measurement results. The results of comparing the actual and predicted values of the TVB-N are shown in Figure 8d, where the comparison of the two different color values shows the performance of the evaluated model. The results show that only a few of the predicted values deviate significantly from the actual values, and the coefficient of determination is 0.96. The comparison between the predicted and actual TVC content is shown in Figure 8e. The results show that the deviation between the predicted results and the actual values is relatively small, and the coefficient of determination is 0.85. Meanwhile, the pH value inside the oyster was also predicted at the same time, but there was a large deviation between the predicted and actual values with a coefficient of determination of 0.67, as shown in Figure 8f. The combined results of the above experiments demonstrated that the use of the TVB-N as an effective quality indicator in the knowledge-based model can effectively regulate the critical control points in the HACCP system of the live oyster cold chain.

### 3.5. Establishment of Live Oyster Cold Chain Record System

Records are the carrier of information and an important means of tracing and reviewing the original data. In order to ensure comprehensive control and management of the oyster cold chain, it is necessary to collect and record the critical environmental factors, the requirements of the different stakeholders, and the potential risks at each stage, as well as upload and update all critical information in time for the implementation of the oyster management plan and create a cold chain management log on the server; identify the stakeholders according to the cold chain business process, collect their needs, and formulate the corresponding quality management plan; and identify stakeholders based on cold chain business processes and gather their requirements to develop a quality management plan accordingly. The monitoring procedure requires the record to include the header, oyster batch, address, reason for deviation, number of deviation products, deviation CCPs point corrective action, and final disposal of affected products.

### 3.6. Evaluation of the Multi-Sensor Traceability System

The evaluation results show that the improved traceability system has an obvious effect on the transportation of live oysters and has been highly appraised by relevant managers, workers, and experts. With a further study and analysis of the relationship between accurate control of transportation processing and transportation distance, they believe that the improved traceability system can effectively reduce transportation costs. In addition, suggestions for further improvement of the established traceability system are given in Table 4, including the use of flexible sensors more suitable for live oysters and the miniaturization and flexibility of related hardware, as well as the development of more functions on the application side, such as optimization suggestions for transport management recommendations. The recommendations of these reviewers have guiding implications for improving traceability systems to achieve more stable and accurate monitoring of pressure levels and management of waterless transport.

## 4. Conclusions

In this study, a flexible wireless sensor network was designed and developed to continuously monitor and record the dynamic fluctuations of the microenvironment (temperature and humidity) and physiological information (shell strength and heart rate). The flexible wireless sensor network can collect the physiological signals of oysters without damage and has the potential to replace the traditional rigid lossy measurement. Live oysters in the cold chain were modeled and evaluated based on flexible wireless sensor networks (F-WSNs) and a knowledge-based HACCP system. This will improve the effectiveness of the HACCP quality control scheme and improve the transparency and traceability of the live oyster cold chain. The HACCP quality control plan of the oyster cold chain was tracked and established. The knowledge model of oyster quality and survival rate based on physiological and environmental signals was established by an artificial neural network. The results showed that the method could effectively evaluate the quality and living condition of oysters.

In addition, the dynamic capture of physiological signals is of great significance to improve quality and survival rates and also provides a reliable and effective research basis for further exploring the relationship between physiological signals and quality. The system evaluation shows that the implementation of HACCP quality control based on a flexible wireless sensor network, and knowledge can improve the quality of raw oysters and improve the efficiency of cold chain management.

## Figures and Tables

**Figure 1 foods-13-00167-f001:**
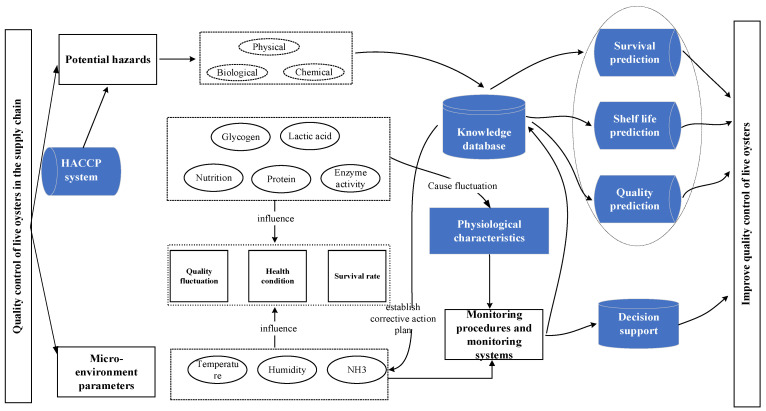
Conceptual framework for quality control of live oyster transport.

**Figure 2 foods-13-00167-f002:**
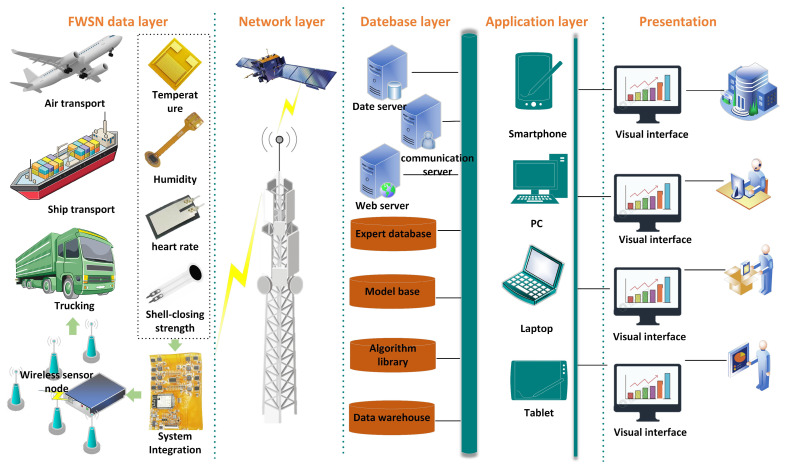
Flexible wireless sensor network (F-WSN) system architecture for live oyster cold chain.

**Figure 3 foods-13-00167-f003:**
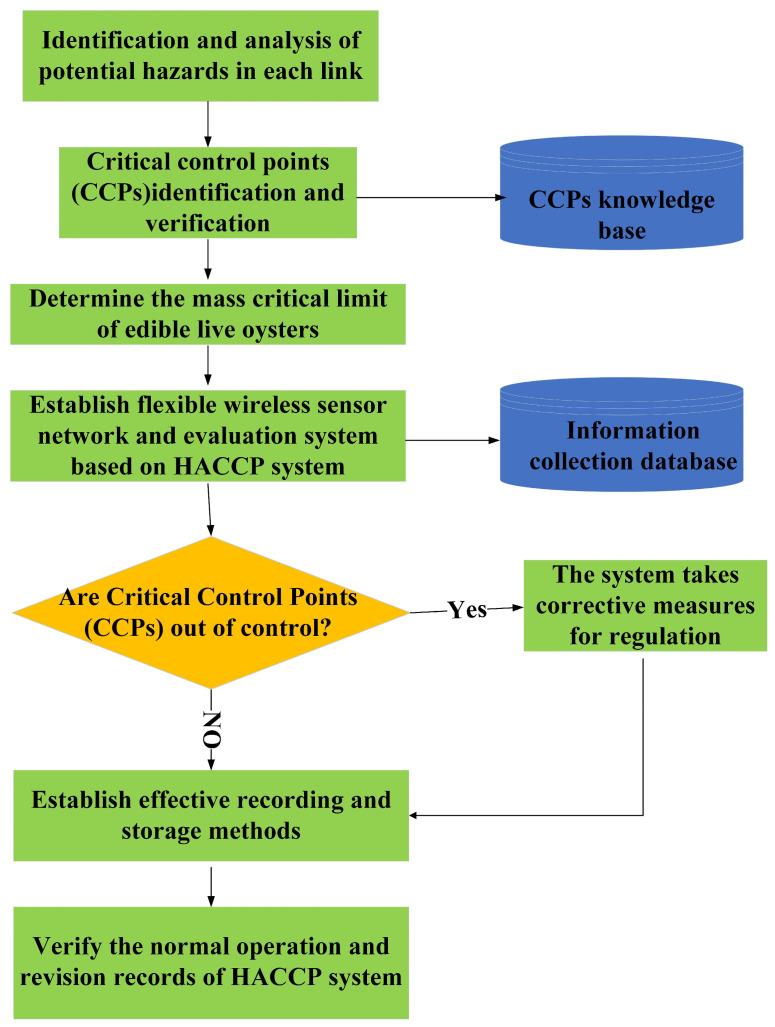
Live oyster cold chain HACCP quality control plan process.

**Figure 4 foods-13-00167-f004:**
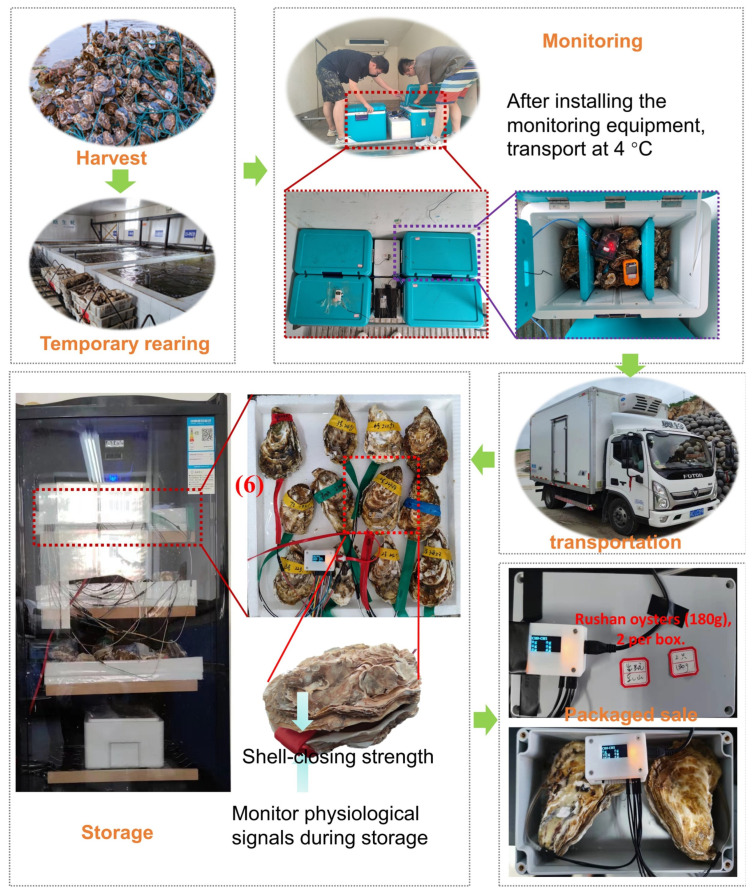
Live oyster cold chain business processes in real scenarios.

**Figure 5 foods-13-00167-f005:**
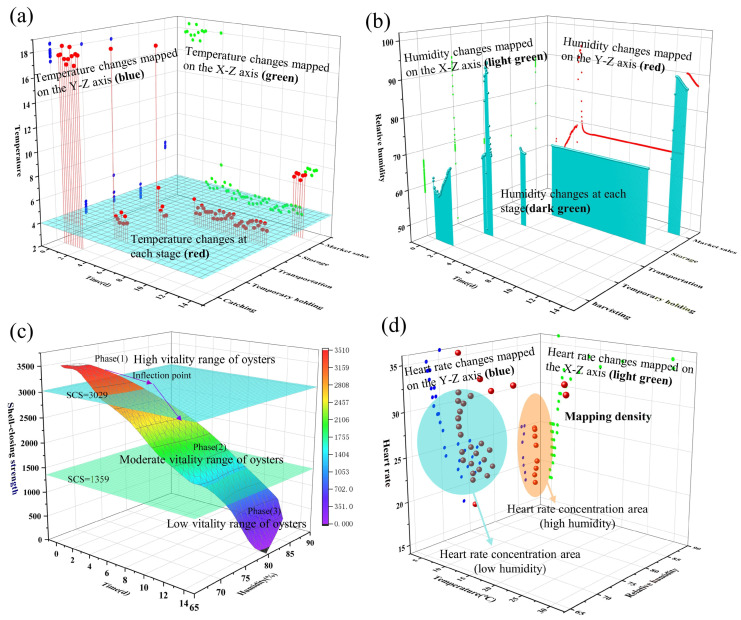
Physiological and environmental signal acquisition at critical control points. (**a**) Changes in temperature at different stages; (**b**) Changes in humidity at different stages; (**c**) The shell-closing strength (SCS) varies over time; (**d**) Changes in heart rate (HR) at different temperatures and humidity.

**Figure 6 foods-13-00167-f006:**
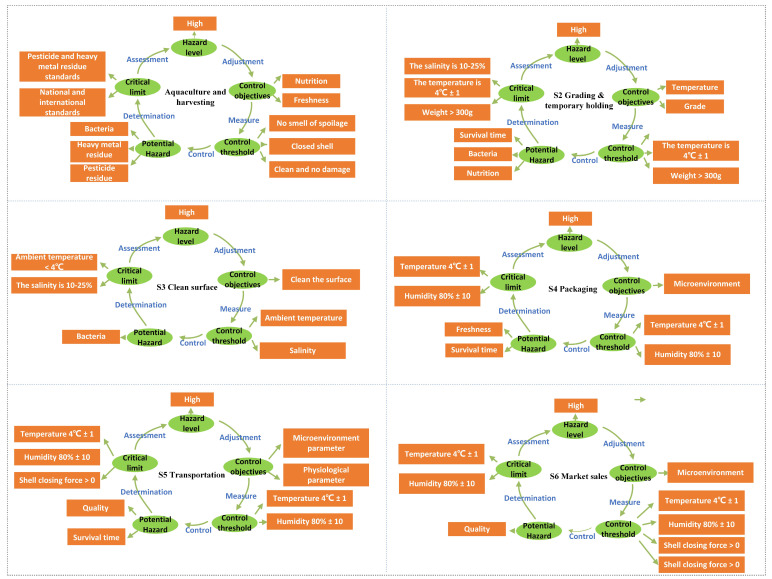
Knowledge-based cold chain critical control point adjustment plan.

**Figure 7 foods-13-00167-f007:**
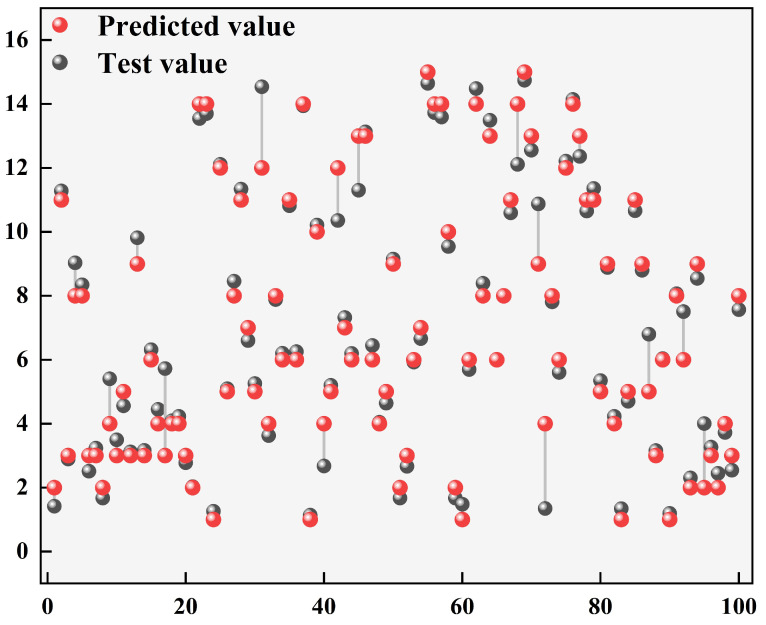
Prediction results of oyster vitality state based on the knowledge model.

**Figure 8 foods-13-00167-f008:**
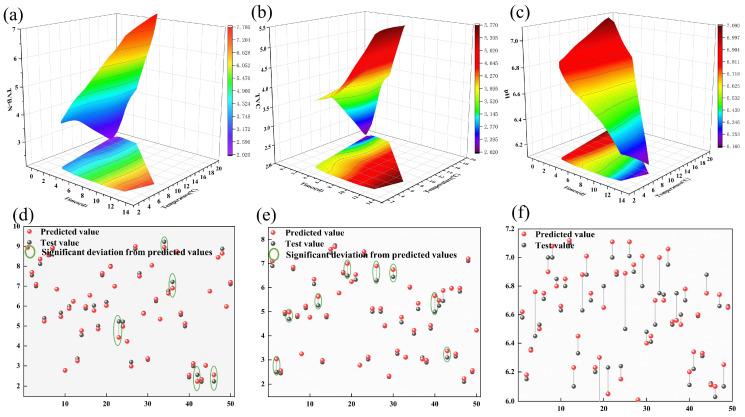
Evaluation results of oyster cold chain quality based on the knowledge model. (**a**) The trend of total volatile base nitrogen (TVB-N) variation; (**b**) The trend of changes in total viable count (TVC); (**c**) The trend of pH value change; (**d**) Prediction of total volatile base nitrogen (TVB-N); (**e**) Prediction of total viable count (TVC); (**f**) Prediction of pH value.

**Table 1 foods-13-00167-t001:** The oyster cold chain quality management planning and implementation scheme.

Cold Chain Process	HA	CM	CCPs	CL	CA	RE
S1 Live oyster farming and harvesting.	Drug, heavy metal residue	Reasonable use of pesticides and feed by rules	Yes	Drug and heavy metal residue standards	Strictly control the usage amount; water quality commissioning testing and assessment	Yes
S2 Grading and temporary holding	Different ability to resist stress	Graded by weight	Yes	Weight > 300 g	The weight of the substandard close sales or continue to breed	Yes
	Cause oyster death	Strictly control water temperature and salinity	Yes	Temperature 2–4 °C	Use special equipment to adjust the temperature	Yes
S3 Clean surface	Affect the quality and appearance	Machine cleaning	No	-	-	Yes
S4 Packaging	Affect the shelf-life and quality	Special transport box	No	Temperature 4 ± 1 °C	Automatic feedback for temperature adjustment	Yes
			No	Humidity 80% ± 10	Automatic feedback to adjust humidity	Yes
			No	Physiological index effective	-	Yes
S5 Transportation	Quality fluctuation and survival rate decrease	Adjust environmental parameters and transportation time	Yes	The model automatically calculates the results	Sell nearby or shorten shipping times	Yes
S6 Storage	Decreased viability and quality decay	Adjust environmental parameters	Yes	70%	Automatic feedback for temperature and humidity adjustment	Yes
S7 Market sales	Quality decay (survivability descent, change of sensory)	Strictly obey the operation rules of live transportation above	Yes	Quality prediction result	Timely recall of products	Yes

**Table 2 foods-13-00167-t002:** Temperature distribution analysis of live oysters at different cold chain stages.

Cold Chain Stage	Maximum	Minimum	Mean	Standard Deviation	CCP	Duration	Control Measures
S1 Aquaculture and catching	18.15	16.9	18.78	0.15	Yes	2 d	Reduce time
S2 Grading and temporary holding	18.11	3.96	4.57	0.09	Yes	2 d	Adjust the temperature
S3 Clean surface	4.09	3.94	4.02	0.01	No	30 min	-
S4 Packaging	4.08	4.01	4.03	0.007	No	2 h	Reduce time
S5 Transportation	17.3	3.89	4.66	1.1	Yes	13 h	Precooling
S6 Storage	4.01	3.96	4.0	0.004	Yes	8 d	Precooling
S7 Market sales	7.56	7.33	7.43	0.03	Yes	1 d	Lower the temperature

**Table 3 foods-13-00167-t003:** Trends in oyster SCS over time (mean ± SD). CV means the coefficient of variation, while different small letters indicate significant differences between time intervals within each group (*p* < 0.05).

Grades	Storage Time	SCS (g)	CV
250 g ± 10 g	0	3493 ± 59.96	0.017
	3	3452 ± 45.87	0.013
	6	2825 ± 113.21	0.040
	9	1112 ± 66.72	0.060
	12	402 ± 12.84	0.032

**Table 4 foods-13-00167-t004:** The subject evaluation before and after application of a live oyster cold chain HACCP system.

ID (Interviewer)	Step	Take Measures	Effect Evaluation	Quality Evaluation	VE	Overall Score	Suggestions
Before application of HAACP system	S1	-	The initial quality and heavy metal content of oysters cannot be guaranteed	poor	No	3	-
	S2	No grading	Poor economic performance; no quality guarantee	No guarantee	No	3	-
	S3	Simply rinse the surface of the oyster shell with water	No cleaning record, no adjustment and control of cleaning environment	Raise slightly	No	3	-
	S4	Foam box with ice pack	Low cost; high mortality rate; poor economic performance	Poor	No	2	-
	S5	Common cold chain transport truck	Without microenvironment control, product quality shelf-life, nutrition and quality cannot be guaranteed	Poor	No	4	-
	S6	Direct sales	Degradation of quality	Poor	No	2	-
After the application of HAACP system	S1	Commissioned tests for water quality and heavy metals; live oysters are healthy, fresh, undamaged, and odorless	Effectively ensure the initial quality of oysters	Valid guarantee	Yes	5	To establish a complete oyster initial status assessment system
	S2	Effective grading of oysters; monitor temperature, salinity, and dissolved oxygen	Effective removal of surface dirt	Valid guarantee	Yes	5	Reduce the hardware volume to explore the overall flexible development
	S3	Precise control of ambient temperature and water salinity	Effective and precise environmental regulation	Valid guarantee	Yes	5	Optimized packaging process
	S4	Special oyster transport box; precise control of temperature and humidity	Effective control of microenvironment to ensure oyster quality and survival rate	Valid guarantee	Yes	5	To explore more physiological signal indicators of oyster and establish better model evaluation
	S5	Precise control of temperature, humidity	Effective control of microenvironment to ensure oyster quality and survival rate	Valid guarantee	Yes	5	To explore more physiological signal indicators of oyster and establish better model evaluation
	S6	Precise control of temperature and humidity in special retail box	Effective control of microenvironment to ensure oyster quality and survival rate	Valid guarantee	Yes	4	To explore flexible intelligent packaging development

## Data Availability

Data belong to the research group; data are confidential and cannot be shared publicly; if necessary, you can send email request.

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
