# Peer review of "Flexible Sensing Enabled Nondestructive Detection on Viability/Quality of Live Edible Oyster"

_foods, 2024, doi:10.3390/foods13010167_

Round 1

Reviewer 1 Report

Comments and Suggestions for Authors

The author proposed a method "Flexible sensing enabled dynamic detection on viability/quality of live edible oyster during cold chain. The topic is interesting and presented well. However, after the comprehensive review, there are many shortcomings in the proposed research work, which need to be addressed, which are;
1. The authors need to rewrite the article in a completely different manner, focusing on the problem statement, significance, and contributions, which are very important to discuss in detail.
2. The title of the paper is confusing, it needs to be meaningful and comprehensive to reflect the theme of the paper.
3. All figures need to be properly cited in the text; moreover, they need to be improved to better quality and must be visible.
4. What is the novelty of the paper, that is not clearly mentioned in the proposed research work?

5. There are many typos and grammatical mistakes that need to be addressed in revision.

6. In the sentence "The oyster cold chain HACCP knowledge domain consists of a set of abstract con- 300 (𝐶) and a set of instances (𝐼) associated with those concepts. The concept set is a 301  collection of all the concepts included in the HACCP plan, 𝐶𝑛 = {𝐶1 , 𝐶2 , . . . , 𝐶𝑖 }; each con- 302 accept contains multiple instance sets, 𝐼𝑛 = {𝐼1 ,𝐼2 , . . . ,𝐼𝑖 }, where 𝑛 = {1,2, . . . , 𝑖}. The HACCP 303" on what criteria the C and I are set.

Comments on the Quality of English Language

Minor changes required.

Author Response

Dear reviewer:

        Hello! Thank you very much for your comments and professional advice. These points contribute to the academic rigor of our article. According to your suggestions and requests, we have made corrections to the revised manuscript. Please refer to the attachment for details.

Reviewer 2 Report

Comments and Suggestions for Authors

The paper has an interesting topic and has detailed information. All the sections were enough and were collected in suitable order. But some factors should be checked;

In Figure 3, the words can’t be read by a reader.

Equation numbers should be written right side.

All the sections of the result section should be supported by references. It should be checked.

The first letter of humidity should be checked in 3.2.2 section.

Author Response

(The authors gave the same response as above.)

Reviewer 3 Report

Comments and Suggestions for Authors

The authors have done a great job, especially in synthesis of results and explanation, it is very clear, so I congratulate the authors and just leave them a few small points

Explain the authors why the standard deviation of table 2 for S2 Grading & temporary holding is higher than S1 Aquaculture & catching, being the mean of the latter almost 5 times higher.

The graphs in Figure 5 should be modified as there are some axes that are not well understood.

The graphs in Figure 8 should be modified as there are some axes that are not well understood.

The conclusions, although very acceptable, I believe that they could be extended with the great work of results exposed.

Author Response

(The authors gave the same response as above.)

Round 2

Reviewer 1 Report

Comments and Suggestions for Authors

The authors address my concern, so i accept the paper for publication.

Comments on the Quality of English Language

N/A